# Control of STING Agonistic/Antagonistic Activity Using Amine-Skeleton-Based c-di-GMP Analogues

**DOI:** 10.3390/ijms23126847

**Published:** 2022-06-20

**Authors:** Yuta Yanase, Genichiro Tsuji, Miki Nakamura, Norihito Shibata, Yosuke Demizu

**Affiliations:** 1National Institute of Health Sciences, 3-25-26 Tonomachi, Kawasaki 210-9501, Japan; w205444b@yokohama-cu.ac.jp (Y.Y.); pvsq1jhc@s.okayama-u.ac.jp (M.N.); n-shibata@nihs.go.jp (N.S.); 2Graduate School of Medical Life Science, Yokohama City University, Yokohama 230-0045, Japan; 3Graduate School of Medicine, Dentistry and Pharmaceutical Sciences, Division of Pharmaceutical Science of Okayama University, 1-1-1 Tsushimanaka, Kita 700-8530, Japan

**Keywords:** STING, cyclic dinucleotide, amines, drug design, agonist, antagonist

## Abstract

Stimulator of Interferon Genes (STING) is a type of endoplasmic reticulum (ER)-membrane receptor. STING is activated by a ligand binding, which leads to an enhancement of the immune-system response. Therefore, a STING ligand can be used to regulate the immune system in therapeutic strategies. However, the natural (or native) STING ligand, cyclic-di-nucleotide (CDN), is unsuitable for pharmaceutical use because of its susceptibility to degradation by enzymes and its low cell-membrane permeability. In this study, we designed and synthesized CDN derivatives by replacing the sugar-phosphodiester moiety, which is responsible for various problems of natural CDNs, with an amine skeleton. As a result, we identified novel STING ligands that activate or inhibit STING. The cyclic ligand **7**, with a cyclic amine structure containing two guanines, was found to have agonistic activity, whereas the linear ligand **12** showed antagonistic activity. In addition, these synthetic ligands were more chemically stable than the natural ligands.

## 1. Introduction

Innate immunity is the first defense mechanism against pathogens. Unlike acquired immunity, innate immunity functions independently of the type of foreign substance, thereby leading to a rapid defense response against pathogens. The recognition of pathogens in innate immunity is mediated by pattern-recognition receptors (PRRs), which detect characteristic molecular patterns of bacteria and viruses (pathogen-associated molecular patterns, PAMP) and signals called damage/danger-associated molecular patterns (DAMP), which are released from injured cells [1]. Recently, cyclic guanosine monophosphate-adenosine monophosphate synthase (cGAS), which is a PRR that recognizes double-stranded DNA (dsDNA) in the cytoplasm, was discovered in mammalian cells [2,3,4,5,6]. cGAS detects cytoplasmic DNA and activates downstream-signaling cascades to stimulate the production of type I interferon and to induce innate immune responses by the cGAS-stimulator of the interferon genes’ (STING)-tank-binding kinase 1 (TBK1) pathway [7]. Thus, cGAS plays an important role in enhancing cytokine production [8].

cGAS recognizes dsDNA in the cytoplasm that is either self-derived (particularly in senescent cells and cancer cells) or exogenous (from bacteria, DNA viruses, and RNA viruses) [2] and produces 2′,3′-cyclic guanosine monophosphate-adenosine monophosphate (2′,3′-cGAMP) [9,10,11,12]. 2′,3′-cGAMP acts as a second messenger that binds to and activates a type of homodimeric endoplasmic reticulum (ER)-membrane receptor, STING [13,14]. In addition, cyclic dinucleotides (CDNs), such as bis-(3′-5′)-cyclic dimeric guanosine monophosphate (c-di-GMP) [15] and bis-(3′-5′)-cyclic dimeric adenosine monophosphate (c-di-AMP), are also unique second messengers produced by bacteria that promote STING-mediated immune activation (Figure 1) [16,17]. Ligand-bound STING migrates from ER to the Golgi apparatus [18] and then undergoes palmitoylation of Cys88/91, resulting in oligomerization [19,20]. The oligomerized STING binds to TBK1 via the carbon-terminal tail (CTT), causing phosphorylation (activation) of TBK1 and activated TBK1 in STING phosphorylates interferon regulatory-factor 3 (IRF3). This series of events promote the dimerization and nuclear translocation of IRF3 [21,22]. In addition, activated STING promotes nuclear-factor kappa B (NF-κB) activation and nuclear translocation via I kappa B kinase (IKK) activation, promoting cytokine production [23]. STING is finally transported to the lysosome for degradation, and STING-mediated signaling converges [24]. STING activation enhances the immune-system response; therefore, it is expected to be utilized in cancer immunotherapy [25,26]. For example, tumor formation was effectively suppressed by combined treatment with STING agonists [27,28,29,30,31,32,33] and adjuvant agents [34,35,36,37,38]. STING activation has also been reported to inhibit viral infections. Since this is an immune-enhancing effect rather than targeting a specific pathogen, it is effective against a wide range of targets, such as human immunodeficiency virus (HIV) and severe acute respiratory syndrome coronavirus 2 (SARS-CoV-2) [39,40,41]. In addition, homeostatic activation of STING causes abnormal immune activation and cytokine production. Thus, STING is a causative protein for autoimmune diseases, such as amyotrophic lateral sclerosis (ALS) and Sjögren’s syndrome, for which there is still no cure [42,43,44]. These observations suggest that STING antagonists [45,46,47] may be potential therapeutic agents for treating diseases related to the homeostatic activation of STING. Therefore, STING has attracted attention as an important drug-discovery target, potentially involved in various diseases. In fact, many studies on STING ligands have been reported in recent years [27,28,29,30,31,32,33,45,46,47]. However, most of the reported agonists were developed to target cancers, and there are few examples of their application in the treatment of infectious diseases [25,26,34,35,36,37,38,39,40,41]. Most antagonists developed are covalent ligands, which raises concerns about specificity, and there are few reports of non-covalent antagonists, many of which have low affinity [45,46,47]. Therefore, there is still a need to develop ligands that target STING because of its diverse roles.

CDN molecules, the natural ligands of STING, have un-druglike properties such as low cell-membrane permeability and are susceptible to inactivation by phosphodiesterase (PDE). These drawbacks must be overcome for CDNs to be used as medicinal drugs. In addition, CDN molecules are difficult to derivatize because of their structure, sugar-phosphate backbones, which are highly polar and reactive moieties, making their chemical synthesis challenging. Most of the currently reported CDN derivatives are limited to those with a phosphodiester skeleton, which are prone to degradation by PDE [48,49,50,51,52,53]. Although several research groups have reported CDN derivatives where the phosphodiester-sugar backbone is replaced with a different linker structure [54,55,56], there are no examples of CDN-based STING ligands with substitution of the entire sugar-phosphodiester backbone. In this study, we designed new CDN mimetics with the phosphodiester skeleton replaced by an amine skeleton, which is easy to synthesize and derivatize. We identified that the amine-skeleton-based molecules activate or inhibit STING. That is, the cyclic ligand with a cyclic amine structure containing two guanines was found to have agonistic activity, whereas the linear ligand showed antagonistic activity.

## 2. Results and Discussion

### 2.1. Design and Synthesis of Compounds

X-ray analysis of c-di-GMP and STING showed that the two guanine moieties of c-di-GMP interact strongly with Tyr164 of STING, via hydrogen bonds and π–π stacking interactions [57]. In contrast, the contribution of the sugar-phosphodiester moiety of c-di-GMP to the interaction with STING is smaller than those of the guanine moieties. Thus, we hypothesized that CDN derivatives with an improved membrane permeability and chemical stability against PDE can be developed, by replacing the sugar-phosphodiester moiety of c-di-GMP with other suitable skeletons.

In the new molecular design, cyclic amines were selected to construct the c-di-GMP mimics. Polyamines are readily available to construct a variety of cyclic structures. In addition to the cyclic analogues, linear-amine derivatives were also synthesized. Each derivative was prepared by introducing the guanine moieties into the cyclic and linear-amine backbones via a condensation reaction (Figure 1 and Figure 2) [58].

### 2.2. IFN-β Induction/Inhibition Activity of CDN Derivatives with an Amine Skeleton

The interaction of the synthesized CDN derivatives with STING was evaluated by a dual-luciferase reporter-gene assay. HEK293T cells, which are reported not to express STING, were employed for this assay [2]. Initially, we evaluated whether a STING-dependent increase in IFN-β gene expression is observed by transient transfection with the gene encoding STING (Appendix A).

An increase in fluorescence intensity was observed by adding cGAMP, confirming that our evaluation system can assess STING-dependent interferon induction. The result showed that compound **7** with a cyclic skeleton exhibited STING-dependent interferon-induction activity, although this was weaker than with cGAMP (Figure 2 and Appendix A). Interestingly, the results also suggested that compound **12** with a linear skeleton displayed antagonistic activity (Figure 3). 

The observed reversal of activity between cyclic compound **7** and linear compound **12** might be attributed to the flexibility of the linker structure. STING is known to change its conformation into several structures upon ligand binding [59], and those that form the same conformational oligomerization to induce subsequent signaling [2]. Moreover, the activity of STING ligands has been reported to be higher in compounds with high rigidity [60], suggesting that the binding of rigid ligands contributes to the activation of STING by fixing the conformation of the protein. For further investigation, we synthesized several linear compounds and evaluated their IFN-β gene-induction activities. That results also showed that linear-amine-based compounds tend to exhibit antagonist activity (Appendix A). 

In this assay, cGAMP requires digitonin treatment to permeabilize the cell membrane, whereas amine-skeleton-based compound **7** showed relatively higher activity than cGAMP, even without digitonin treatment. When cells were not permeabilized with digitonin, the activity of cGAMP was markedly reduced to about 1/178; however, the decrease in the activity of compound **7** with the amine skeleton was relatively small, at about 1/27 (Appendix A). Thus, replacing the amine skeleton appears to eliminate the negative charge and improve the cell-membrane permeability, when compared with CDNs with the sugar-phosphodiester moiety.

### 2.3. Stability of CDN Derivatives against Nuclease

The phosphodiester bond of CDNs has been reported to be digested by hydrolytic enzymes in vivo. Thus, to evaluate the chemical stability of the synthesized compounds, CDN derivatives were treated with nuclease P1 (NP1), which cleaves phosphodiester bonds to yield linear 5′-monophosphate nucleotides. The digested products in the reaction mixture were confirmed by HPLC analysis. In the presence of NP1, rapid digestion of CDNs was observed (Appendix A). In contrast, no digestion was observed for the amine-skeleton cyclic compound **7** without the sugar-phosphate backbone (Appendix A). This result indicates that it is possible to replace the sugar-phosphate moiety with an amine moiety, to obtain stability against at least NP1.

### 2.4. Docking Study between STING and the Binding Ligands

Finally, we performed docking simulations to analyze the binding mode of compounds **7** and **12**, which showed agonist and antagonist activities toward STING, respectively. Docking simulations were performed for compounds **7** and **12** using the crystal structures of the active (PDB ID: 4LOH) and inactive (PDB ID: 6MXE) forms of STING, respectively. Since it has been reported that two molecules of cpd. **18**, an existing STING inhibitor, bind to the ligand-binding domain (LBD) and inhibit the activity of STING [46], we simulated compound **12** with two molecules in addition to the simulation with a single molecule. The results suggest that compounds **7** and **12** bind to the ligand-binding site of STING in a structure similar to that of the ligand in their respective crystal structures (Figure 4 and Appendix A). In particular, the results indicated that compound **7** maintains π–π stacking interactions between the guanine base and Tyr164, which is thought to be important for the activation of STING (Figure 4A and Appendix A). Docking simulations with two molecules of compound **12** suggest that compound **12** interacts with the broader surface of STING via a hydrogen bond with Pro264, as in cpd. **18**, indicating that two molecules of compound **12** can bind to STING (Figure 4C and Appendix A).

## 3. Materials and Methods

All reagents were obtained from commercial suppliers and were used as received, unless otherwise noted. 

### 3.1. Synthesis and Characterization of Compounds

The synthesis procedure and analytical data for the synthesized compounds are provided in the Appendix A.

### 3.2. Dual Luciferase-Reporter-Based Biological Evaluation in HEK293T Cells 

HEK293T cells [61] were maintained in DMEM (Sigma-Aldrich, St. Louis, MO, USA) and were supplemented with 10% FBS (Sigma-Aldrich) and penicillin-streptomycin (Nacalai Tesque, Kyoto, Japan). The expression plasmid for human STING (Flag-STING) was constructed by inserting a fragment coding for STING into the p3×Flag-CMV10 (Flag-Empty; Sigma-Aldrich). The fragment was generated from a PCR product. HEK293T cells were plated in 24-well plates at 1 × 10^5^ cells/well. The next day, the cells were transfected together with Flag-STING or Flag-Empty as well as IFN-β-firefly luciferase (#102597, Addgene, Watertown, MA, USA) [62] and SV40-Renilla luciferase (#E223A, Promega, Madison, WI, USA)-reporter constructs using Lipofectamine LTX (Thermo Fisher Scientific, Tokyo, Japan). Compounds were mixed with a 100× volume of digitonin permeabilization solution (50 mM HEPES, pH 7.0, 100 mM KCl, 3 mM MgCl_2_, 0.1 mM DTT, 85 mM sucrose, 0.2% BSA, 1 mM ATP, 0.1 mM GTP, 10 μg/mL digitonin) or DMEM medium. The medium was aspirated from the cells and replaced with 200 μL for each sample mixture. Cells were incubated for 30 min at 37 °C. Wells were again aspirated, and fresh DMEM medium (FBS: +, P/S: –, 500 μL/well) was added. Following stimulation for 6 h with the compounds, the cells were lysed in passive lysis buffer (Promega) for 15 min at 37 °C. The cell lysates were incubated with firefly luciferase substrate (Promega) and the Renilla luciferase-substrate coelenterazine (Promega), and the luminescence was measured using a Wallac ARVO Sx 1420 Multilabel Counter (PerkinElmer Japan, Chiba, Japan). The relative IFN-β expression was calculated as firefly luminescence relative to Renilla luminescence.

### 3.3. Stability Test against an Exonuclease

cGAMP (2′3′-cGAMP; InvivoGene, San Diego, CA, USA) and CDN derivatives (1 µg) were incubated in a solution (20 μL) containing the enzyme (New England Biolabs, Ipswich, MA, USA; 2.5-mU NP1 in 50 mM acetate buffer containing 1.25 mM Tris-HCl, 2.5 mM NaCl, 50 nM ZnCl_2_ (pH 5.5)) or 50 mM acetate buffer (negative control) for 15 min at 37 °C (water bath). The reaction was terminated by heat inactivation at 75 °C for 10 min. Ten microliters of each aliquot were injected directly into the HPLC (column: CAPCELL PAK MG-II [C18, 4.6 × 250 mm, 5 μm] (Osaka soda, Osaka, Japan); flow rate 0.6 mL/min; detection at 254 nm) for analysis. TEAA buffer (5 mM, pH 7.0) (solvent A) and acetonitrile (solvent B) were used as the mobile phase. The gradient was set as follows: 0–30 min: 0% B to 20% B; 30–35 min: 100% B; 35–40 min: 0% B (for c-di-GMP); or 0–30 min: 5% B to 50% B; 30–35 min: 100% B; 35–40 min: 5% B (for cGAMP and CDN derivatives).

### 3.4. Molecular Docking

Possible interactions between amine-skeleton CDN derivatives and STING (PDB IDs: 4LOH, 6MXE) were examined by performing docking studies using the Molecular Operating Environment (MOE) 2020.0901. The binding site of the protein was defined as the native ligand in the X-ray structure. Docking simulations of amine-skeleton CDN derivatives bound to STING were carried out using the standard protocols of general docking. The docking workflow followed an “induced fit” protocol that allowed the side chains of the receptor pocket to move according to the conformation of the ligand, and the positions were constrained. The weight used to lock the side-chain atoms in their original positions was 10. All docking poses were first ranked by the London dG score; then, a force field refinement was performed on the top 100 poses, followed by s rescoring of GBVI/WSA dG. The Amber14:EHT [63] force field was used for calculating the conformations.

## 4. Conclusions

In this study, we synthesized CDN analogues, in which the phosphate backbone is replaced with the amine moiety, as new STING ligands and found them to be stable against nucleases. In particular, cyclic-amine compound **7** and linear-amine compound **9** were found to have agonistic and antagonistic activities, respectively. Furthermore, docking simulations suggest that the agonistic (for **7**) and antagonistic (for **9**) actions observed in those molecules are due to differences in their interactions with STING. Although the activities of the amine-skeleton-based CDN molecules found in this study are still lower than that of cGAMP, it is possible to develop more active ligands by improving the physical properties, such as cell-membrane permeability and chemical stability. The derivatization of further CDN ligands that target STING and detailed studies of their mechanism of activity are currently underway.

## Data Availability

The data presented in this study are available on request from the corresponding author.

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
