# Peer review of "Control of STING Agonistic/Antagonistic Activity Using Amine-Skeleton-Based c-di-GMP Analogues"

_ijms, 2022, doi:10.3390/ijms23126847_

Round 1
Reviewer 1 Report
The work conducted by Demizu’s team deals with the synthesis and characterisation of ligands binding to STING for their applications of the regulation of immune system. Several cyclic-di-nucleotides have been prepared and the sugar-phosphodiester moiety has been replaced by an amine skeleton. The synthetic procedure was described and the fully characterisation was achieved by 1H NMR spectroscopy. It is an interesting pathway which can be useful for the community. The immune response was evaluated by a dual luciferase assay showing different behaviours depending on the ligands, either as agonist or antagonist of STING. By replacing the sugar-phosphate backbone, the authors increase the drug-like properties of the ligands as well as increase the cell-uptake. In addition, both ligands are more stable to nuclease degradation due probably to the steric hindrance of the proteins to embrace the DNA to end up cleaving the nucleic acid. They completed by docking both molecules to STING and correlate the activity to the matching between the binding pocket and the ligand structures. The work lacks of something important such as final section with the main conclusions which must be included before publishing. Moreover, the authors shall hypothesise the different immune response of the ligands depending on the STING binding or mechanism of action since they have not explained it.
Author Response
Answer :
According to the reviewer’s comment, the authors added “3. Conclusion” section in the revised manuscript.
3. Conclusion
In this study, we synthesized CDN analogues in which the phosphate backbone is replaced with the amine moiety as new STING ligands, and found them to be stable against nucleases. In particular, cyclic amine compound 7 and linear amine compound 9 were found to have agonistic and antagonistic activities, respectively. Furthermore, docking simulations suggest that the agonistic (for 7) and antagonistic (for 9) actions observed in those molecules are due to differences in their interactions with STING. Although the activities of the amine-skeleton based CDN molecules found in this study are still lower than that of cGAMP, it is possible to develop more active ligands by improving physical properties, such as cell-membrane permeability and chemical stability. The derivatization of further CDN ligands that target STING and detailed studies of their mechanism of activity are currently underway.
It is also about the possibility of the different immune response by compounds developed in this study, as the reviewer mentioned, although many of the reported papers on STING ligands confirm the dependence on the cGAS-STING pathway, the possibility that IFN-beta is produced as a result of immune system regulation via other targets (e.g. TLR3, RIG-I etc.) cannot be ruled out. It is possible that the molecules developed in this study also affect INF-beta production by acting on targets other than STING. However, at least in their involvement in the IFN-beta expression pathway, the experimental data (Fig. 2 and Fig. S2) suggest that their contribution is quite low. We would like to investigate on the possibility of contributing to other targets.
Reviewer 2 Report
The manuscript is well written and presented. The work reported involves the use of cyclic-dinucleotides (CDNs) as ligands of the drug target, Stimulator of Interferon Genes, Sting. The phosphodiester bond in CDNs impedes membrane permeability and is a ready target for endogenous phosphodiesterases. Novel compounds replacing the sugar-phosphate moieties of CDN with both cyclic and linear amine structures. These modifications then allow the binding of the dinucleotides to the target.
One issue with the research plan is that the stability study examined only resistance to a phospodiesterase, nuclease P1. This only serves to confirm the obvious, that the compounds are not NP1 substrates. It would have been more interesting to examine a wider range including stability in a cell lysate. For the nuclease test, a CDN could have been included as a control.
The only text suggestion is on line 98. There are many kinds of X-ray analysis. It would be more readable to say, "Crystal structure analysis."
Author Response
Ansewer :
The authors totally agree the reviewer’s opinion. To conclude the superiority of stability of compounds developed in this study, it is necessary to evaluate other enzymes and other enzymes. However, considering the deadline in revision, it is difficult to establish proper evaluation systems, and we would like to discuss this issue in the near future.
Reviewer 3 Report
The manuscript is very well written and documented. The only thing I have reservations about is that I would like to see all NMR spectra (including two-dimensional spectra) and mass chromatograms.
Author Response
Ansewer :
In this study, it was difficult to obtain 13C NMR spectra for compounds that were synthesized in small quantities, therefore, only 1H NMR spectra are provided for them. The authors appended 2D NMR (HMBC) spectra for confirmation of N-alkylating position of compound 30 in supporting information. Also, MS chromatograms of key compounds (X, Y and Z) were added in supporting information.
Round 2
Reviewer 1 Report
The problem has been fixed by the authors and the manuscript is good for publication.